# Surface- and Tip-Enhanced Raman Scattering by CdSe Nanocrystals on Plasmonic Substrates

**DOI:** 10.3390/nano12132197

**Published:** 2022-06-26

**Authors:** Ilya A. Milekhin, Alexander G. Milekhin, Dietrich R. T. Zahn

**Affiliations:** 1Semiconductor Physics, Chemnitz University of Technology, D-09107 Chemnitz, Germany; zahn@physik.tu-chemnitz.de; 2Center for Materials, Architectures and Integration of Nanomembranes (MAIN), Chemnitz University of Technology, D-09107 Chemnitz, Germany; 3Novosibirsk State University, 630090 Novosibirsk, Russia; milekhin@isp.nsc.ru; 4A.V. Rzhanov Institute of Semiconductor Physics, Siberian Branch of Russian Academy of Sciences, 630090 Novosibirsk, Russia

**Keywords:** CdSe nanocrystals, surface enhanced Raman scattering, SERS, tip enhanced Raman scattering, TERS, resonant TERS

## Abstract

This work presents an overview of the latest results and new data on the optical response from spherical CdSe nanocrystals (NCs) obtained using surface-enhanced Raman scattering (SERS) and tip-enhanced Raman scattering (TERS). SERS is based on the enhancement of the phonon response from nanoobjects such as molecules or inorganic nanostructures placed on metal nanostructured substrates with a localized surface plasmon resonance (LSPR). A drastic SERS enhancement for optical phonons in semiconductor nanostructures can be achieved by a proper choice of the plasmonic substrate, for which the LSPR energy coincides with the laser excitation energy. The resonant enhancement of the optical response makes it possible to detect mono- and submonolayer coatings of CdSe NCs. The combination of Raman scattering with atomic force microscopy (AFM) using a metallized probe represents the basis of TERS from semiconductor nanostructures and makes it possible to investigate their phonon properties with nanoscale spatial resolution. Gap-mode TERS provides further enhancement of Raman scattering by optical phonon modes of CdSe NCs with nanometer spatial resolution due to the highly localized electric field in the gap between the metal AFM tip and a plasmonic substrate and opens new pathways for the optical characterization of single semiconductor nanostructures and for revealing details of their phonon spectrum at the nanometer scale.

## 1. Introduction

A general trend of modern (opto-)electronics is the steady decrease in the size of active elements in devices. At characteristic element sizes of the order of 1–10 nm, quantum confinement effects become noticeable [1,2], which markedly change the electronic and vibrational spectra of the materials. Note that it is also possible to solve the inverse problem and synthesize nanomaterials with defined electronic and vibrational properties. Semiconductor nanocrystals (NCs) are an ideal model object, the optical properties of which can be varied over a wide range via changing the NC size. In particular, the most studied CdSe NCs with sizes in the range of 2 to 6 nm have interband transition energies varying from the blue to the red spectral range depending on the NC size, thus covering the entire visible spectral range [3]. Raman scattering by longitudinal optical (LO) phonons with a frequency of about 210 cm^−1^ dominate the Raman spectra of bulk cubic CdSe crystals [4,5]. Reducing the crystal size in three dimensions to the nanometric range leads to the quantum confinement effects. Materials possessing a negative optical phonon dispersion reveal a Raman shift of the LO and transverse optical (TO) modes towards lower frequency (around several cm^−1^), accompanied by an asymmetric peak broadening [6,7,8]. In general, CdSe NCs exhibit diverse electrical and optical properties [9,10], which depend on the NC size [11] and morphology [12], and are of both fundamental and technological interest. Owing to the ability to smoothly tune the absorption/emission energy, semiconductor NCs are promising for usage in (opto-)electronic devices, e.g., light-emitting devices, solar cells, memory devices, or as optical markers in biology and medicine [13]. The current level of nanotechnology development makes it possible to create semiconductor NCs with controlled structural parameters, including size, shape, phase, and chemical compositions [14] for optoelectronic devices with superior characteristics, e.g., for super-saturated color TV displays [15]. At the same time, the correlations between the structural, optical, electronic, and vibrational properties of NCs are still the subject of intense research. In this respect optical spectroscopic methods traditionally used for studying bulk materials, thin films, and nanostructures such as Raman scattering, IR absorption, and photoluminescence are restricted by the diffraction limit. When studying nanostructures, they usually deal with ensembles of nanoobjects with a dispersion in size, shape, and composition. As a result, the spectral response of nanostructure ensembles turns out to provide limited information, since it is averaged over a significant number of NCs, and the contribution of a single NC is below the detection threshold or smeared out. Therefore, experimental investigations of the phonon spectrum of single NCs are rather scarce.

This is one of the main reasons why recently the methods of plasmon-enhanced Raman spectroscopy (initially SERS) have been intensively developed to enhance the signal from individual NCs employing the spatially confined enhancement of the electromagnetic field near metal nanostructures under excitation near the LSPR in the visible or IR range [16]. Typically, metallic nanostructures usually composed of Au, Ag, Cu, or Al with a size of 10–100 nm have LSPR energies in the visible spectral range [17]. The Raman scattering intensity of semiconductor NCs placed near metal nanoclusters increases manifold due to the enhancement of the local field. This makes it possible to study the phonon spectrum by SERS of low-density NC coverages.

At present, the phonon properties of CdSe based semiconductor nanostructures, such as either bare CdSe NCs or nanostructures with more complex morphology including core-shell NCs and [18,19,20,21], and core/shell and core-crown nanoplatelets [22,23,24] are actively studied by both conventional Raman and SERS experiments. Due to the fact that the LSPR energy can vary depending on the structural parameters of the plasmonic nanostructures, it is possible to achieve a coincidence of the LSPR energy and the excitation energy, and by placing semiconductor NCs onto specially designed arrays of, for example, Au nanostructures with regularly varied size to enhance resonantly the SERS signal. The significant SERS enhancement factor EF_SERS_ scales with the 4th order of the magnitude of the local electric field E, i.e., EF_SERS_~E^4^ [25]. It is, thus, possible to determine experimentally the dependence of the SERS enhancement factor for NC phonons on the structural parameters of the plasmonic nanostructures and to determine the optimal conditions for observing enhanced SERS spectra.

SERS was successfully used to study the structural evolution of the interfaces in colloidal CdSe/CdS and CdSe/Cd_0.5_Zn_0.5_S core/shell NCs interacting with Ag plasmonic substrates [20]. Hybridization of Au nanowires with CdSe/ZnS core/shell NCs provides an overlapping of the surface plasmon band and the NCs absorption range allowing localized and surface optical phonon modes to be studied in both the CdSe core and the ZnS shell [19]. The LO phonons of a monolayer of CdSe NCs deposited onto a commercial gold SERS substrate (Klarite) can be detected by SERS at 208 cm^−1^ [18]. With the choice of the excitation wavelength (514, 633, or 785 nm), the condition for resonant SERS can be achieved and pave the way for using NCs as SERS markers [18].

Apparently, Raman spectroscopy including SERS is restricted in spatial resolution by the diffraction limit, which amounts to approximately a half of excitation wavelength. To overcome the diffraction limit, tip enhanced Raman scattering (TERS) has been developed [26]. TERS is the near-field technique, which combines atomic force or scanning tunneling microscopy and Raman spectroscopy [26]. TERS can be considered as a limiting case of SERS, where the metallized apex of the scanning probe tip acts as the plasmonic nanostructure allowing spectroscopic nanoscale mapping of nanoobjects. At the tip of the metallized scanning probe, a high electric field or hot spot is generated under laser radiation. Besides the electromagnetic enhancement at the tip, chemical enhancement due to charge redistribution can also contribute to the TERS enhancement of phonon modes in nanostructures [27]. Typically, the local TERS enhancement can reach values of 10^7^ for various organic materials, carbon nanotubes [28,29,30,31], one-dimensional carbyne [30], and graphene [30,32].

Here we provide an overview of our recent experimental results devoted to SERS and TERS studies of CdSe NCs with various concentrations from a NC monolayer down to a single NC.

## 2. Materials and Methods

### 2.1. Fabrication and Characterization of Plasmonic Substrates

Periodic plasmonic substrates consisting of Au nanodisks arrays (Figure 1a–f), nanodimers arrays (Figure 2a), and single nanodimers (Figure 2b,c) were fabricated on Si(001) substrates with natural or 77 nm thick silicon oxide layers by electron beam lithography (Raith−150, RAITH Nanofabrication, Dortmund, Germany), as described in detail elsewhere [33]. The final plasmonic substrates consist of Au nanodisks arrays (Figure 1a–f), Au nanodisk dimer arrays (Figure 2a) [33,34] or single Au dimers (Figure 2b,c) with different structural parameters, namely, diameter of Au nanodisks (30–150 nm), distance between nanodisks (period or pitch size) of 130, 150, 200, and 250 nm, and a height of 50 nm. To verify the structural parameters of nanodisk and dimer arrays, scanning electron microscopy (SEM, Raith−150, RAITH Nanofabrication, Dortmund, Germany) was used.

The nanolithography process employed allows fabricating nanodisks and dimers with a high accuracy of about ±5 nm.

Another type of plasmonic substrate used in SERS experiments called Klarite^®^ structure consists of arrays of inverted Si pyramids as shown in AFM image Figure 3a. They were available as commercially plasmonic substrates especially developed for SERS experiments. As can be seen from Figure 3b, the surface of the Klarite substrate is homogeneously covered by Au nanoclusters with irregular shapes and sizes of about 50–100 nm [35]. The pyramid base is a square with a size of 1.5 × 1.5 µm^2^, while the pyramid depth is 1 µm. The inverted pyramids act as resonators and provide an electromagnetic field enhancement inside the cavity at the vertices of the pyramids [36].

A typical SEM image of the rectangle area of Klarite structure with a deposited CdSe NC submonolayer is presented in Figure 3b. The SERS and TERS experiments with the Klarite structures were performed for comparison with the efficiency of the SERS and TERS response for nanodisks and nanodimers.

### 2.2. The TERS Tip Preparation

Tips for the TERS experiments were prepared by depositing an Au layer on commercially available AFM tips. Standard Si AFM cantilevers purchased from Tipsnano (Tipsnano, Tallinn, Estonia) with a specified tip radius of <10 nm as shown in Figure 4a were used as the basis for the TERS probes. A titanium 5 nm-thick layer for better adhesion and a gold layer with a nominal thickness in the range of 150–200 nm were thermally evaporated onto the apex of the commercial AFM cantilever [37]. To reveal the morphology of the tip, SEM images (Figure 4a,b) were acquired before and after the metal deposition process. As can be seen from Figure 4b, the metal deposition results in the formation of polycrystalline Au grains on the AFM tip and a single gold nanocluster with a typical size of about 80–90 nm is formed at the tip apex.

### 2.3. Methods of Synthesis and Deposition Processes of CdSe NCs

CdSe NCs were synthesized by colloidal chemistry according to a procedure described earlier [2,38,39,40]. Oleic acid was used as a crystal surface stabilizer. Colloidal CdSe NCs dissolved in a toluene solution with a concentration of 10^−3^ M, mixed with a solution of behenic acid in a molar ratio of 1–3: 1, were then used for homogeneous deposition onto plasmonic substrates with various morphologies (nanodisks, nanodimers, and Klarite substrates) by the Langmuir–Blodgett (LB) method, as described in [41,42]. Mono- and submonolayers of colloidal bare CdSe NCs were formed by varying the number of behenic monolayers as described in detail earlier in [43,44]. After the deposition of CdSe NCs within a matrix of behenic acid, the samples were annealed at 150 °C to remove the organic molecules of behenic acid. Using (high resolution) transmission electron microscopy ((HR)TEM), we found that CdSe NCs have a zincblende crystal structure and an average size of 5–6 nm (Figure 5a,b). As a result, submonolayers of colloidal CdSe NCs were formed on the arrays of plasmonic structures [43,44] (Figure 6a–d). 

The HRTEM images were recorded using TITAN 80–300 TEMs (FEI, Hillsboro, Oregon, United States) equipped with a spherical aberration corrector (Cs). High-resolution images were taken at an accelerating voltage of 300 kV.

### 2.4. Raman Spectroscopy Setups

Conventional Raman and SERS experiments were performed using LabRam or XPlorA Raman spectrometers (HORIBA Jobin Yvon GmbH, Bensheim, Germany), each of which was equipped with an Olympus optical microscope, a diffraction grating of 600 lines/mm, and an electron multiplying CCD detector (EMCCD) providing a spectral resolution of 5 cm^−1^. A solid-state laser with the wavelength of 632.8 nm was used as an excitation source. Objectives (100 × 0.9 NA) provide a size of the laser spot of about 1 μm for both XPlorA and LabRam spectrometers.

For TERS measurements, a commercial combined AFM-Raman system XPlorA / AIST-NT TERS (HORIBA Jobin Yvon GmbH, Bensheim, Germany ) was used in a quasi-backscattering configuration under standard laboratory conditions in the semicontact AFM mode. Excitation and detection of the optical signal were carried out through an objective with a long working distance (100 × 0.7 NA) in a side illumination geometry at an angle of 65° relative to the normal sample surface. Solid-state lasers with wavelengths of 638.2 and 785.3 nm and powers of 1 and 3 mW, respectively, were used for excitation. The side illumination geometry is very effective for the excitation of hybrid plasmon states in the so-called gap-mode for both Au nanodisk arrays or the Klarite structure and the TERS tip. Due to the side illumination, the laser spot on the surface has an ellipsoidal shape, where the minor and major axes of the ellipsoid are about 1 and 1.7 μm, respectively. The accumulation time for one spectrum and the step size for mapping were 0.3 s and 6 nm, respectively, which allowed the recording of TERS images with a high lateral resolution with reasonable acquisition times. The spectral resolution was about 7 and 4 cm^−1^ for lasers with 638.2 and 785.3 nm wavelengths, respectively. Note, that spectral resolution was measured for 50 µm slit and 100 µm hole.

## 3. Results and Discussion

### 3.1. SERS of CdSe NCs on Klarite Substrates

The sensitivity of conventional Raman scattering is insufficient for detecting the Raman response of a single CdSe NC monolayer. Therefore, to study the phonon properties, it is necessary to enhance Raman signal, e.g., in a SERS experiment using plasmonic substrates for the CdSe NC deposition. A typical commercially available Klarite structure was chosen for SERS experiments to study phonons of CdSe NCs. In accordance with [35], the LSPR of the Klarite substrate is located around 700 nm allowing resonant enhancement of SERS response by CdSe NCs with excitation wavelengths of 600–700 nm. It should be noted that to achieve resonant SERS, the excitation energy should match the energy of the NC bandgap and the LSPR energy. Thus, to observe resonant SERS from CdSe NCs deposited on the Klarite substrate, a laser with a wavelength of 638.2 nm (1.97 eV) was used since it is close to the bandgap of the CdSe NCs (610 nm, 2.03 eV) and the LSPR energy of the Klarite substrate. The inverted pyramids of the Klarite structure are resonators that locally enhance the electric field inside the pyramids. The SERS spectrum (Figure 7, red curve) of a CdSe NC monolayer measured on a Klarite substrate reveals the CdSe LO and 2LO phonon features [18], while no phonon scattering from NCs deposited on the flat gold (Figure 7, blue curve) was detected. The SERS enhancement factor of the LO CdSe mode is at least 10 with respect to the noise level as can be estimated from Figure 7.

### 3.2. TERS of CdSe NCs on Klarite Substrate

The TERS mapping was performed for corners of the inverted pyramid as shown in Figure 8a,b. as in accordance with [35], the electric field is enhanced at the top and corners of the pyramid. TERS maps of LO CdSe mode of NCs deposited on a Klarite substrate are shown for low (Figure 8a) and high spatial resolution (Figure 8b) recorded with excitation wavelengths of 785.3 and 638.2 nm, respectively. The optical bandgap of CdSe NCs with sizes of about 5–6 nm is approximately 2.03 ± 0.05 eV corresponding to 610 nm [45]. Therefore, resonant Raman conditions for phonon modes of CdSe NCs are approximately met for 638.2 nm excitation. The brightness of TERS maps in Figure 8a,b correlates with the Raman scattering intensity in the frequency range between 160 and 250 cm^−1^ as shown by the green areas highlighted in Figure 8c,d. As can be seen from Figure 8d, a resonant TERS spectrum measured inside the Klarite structure reveals relatively strong first and second order LO, as well as a SO feature of CdSe (Figure 8d, red spectrum), compared to the Raman spectrum taken from CdSe NCs on flat Au (Figure 8d, black spectrum).

In addition to the observation of a quite strong Rayleigh scattering background when excited with the 785.3 nm laser line, the shape of the LO peak observed at 209 cm^−1^ is also modified (Figure 8c, red spectrum) with respect to 638.2 nm (Figure 8d, red spectrum). For 785.3 nm excitation, a broad peak appears as a combination of SO and LO modes, and the intensity of this broad feature is relatively less intense than for the case of 638.2 nm excitation. It should be noted that the TERS spectra in Figure 8d (black spectrum) measured on the surface of flat gold exhibit a weak feature of the LO mode, while this mode is absent in the Figure 8c (black spectrum), which proves the resonance conditions of the experiment using a wavelength of 638.2 nm [43].

### 3.3. SERS by CdSe NC Mono and Submonolayers Deposited on Au Disks and Dimers

Despite the fact that the commercial Klarite structure allows obtaining SERS and TERS from CdSe NCs, we applied alternative plasmon structures to achieve further improved enhancement of SERS and TERS responses from CdSe NCs.

A comparison of conventional Raman and SERS spectra of a CdSe NC monolayer is shown in Figure 9a. The Raman spectrum measured outside of the area of the plasmonic nanostructures for CdSe NCs on the bare oxidized Si substrate corresponds to the Raman spectrum of the Si substrate (Figure 9a, spectrum 1). No evidence of CdSe phonon modes was detected due to insufficient sensitivity of this conventional Raman measurement towards the small amount of CdSe NCs on the surface. However, the SERS spectrum reveals the LO phonon mode of CdSe NCs near 200 cm^−1^ [46], as well as higher-order overtones (2LO and 3 LO).

To estimate the SERS enhancement, the intensities of the CdSe LO phonon mode measured on nanodisks with a pitch of 200 nm and on a flat silicon surface were compared. While CdSe-related Raman signals from the sample with CdSe NCs on the bare Si surface are absent, the SERS enhancement factor for Au nanocluster samples was estimated taking the Raman intensity of the LO phonon mode in the sample with CdSe NCs on the bare Si surface equal to the noise level. The value obtained in this way is as high as 2 × 10^3^ [41]. The SERS intensity of the LO phonon mode as a function of the nanodisk size for different excitation energies demonstrates a noticeable resonance behavior as shown in Figure 9b. With decreasing excitation energy (increasing wavelength), the maximum of the SERS intensity shifts towards larger diameters of the Au nanodisks (Figure 9b) in accordance with the electromagnetic enhancement mechanism of the SERS effect. To achieve resonant Raman scattering from CdSe NCs, it is necessary to fulfill the condition for the coincidence of the excitation energy and the energy of the NC bandgap. Thus, the required condition for resonant SERS is the coincidence of three energies: the bandgap of the NC, the excitation energy, and the LSPR energy. A more detailed discussion of enhancement mechanisms was presented in ref. [41].

Due to the electromagnetic SERS mechanism, Raman scattering from CdSe NCs deposited on Au dimer arrays should exhibit different intensities of the LO mode depending on the light polarization. The SERS effect is expected to be very strong for NCs placed in the gap between the two nanodisks constituting a dimer due to the increased electric field in this area forming a hot spot [47]. Au dimers should, thus, cause more intense SERS enhancement in the case when the incident and scattered light are polarized along (parallel to) the dimer axis as compared to the polarization perpendicular to the dimer axis. The SERS spectra recorded with polarization perpendicular and parallel to the dimer axis are shown in Figure 9c by blue and red spectra, respectively. Indeed, 1 ML of CdSe NCs deposited on arrays of Au dimers (Figure 9c) shows a SERS intensity ratio for the LO phonon mode of about 25 between parallel and perpendicular polarizations of light.

Using polarization parallel to the dimer axis, SERS was observed for dimers coated with a NC monolayer (Figure 9c). If we take into account that the size of a single CdSe NC is 6 nm, and the laser beam diameter is 1 μm, then we can estimate the number of NCs that contribute to Raman and SERS signals. In this case, the number of NCs in a NC monolayer irradiated by the laser is about 3 × 10^4^. However, the signal from the monolayer outside the plasmonic structures is negligible. For SERS, we take into account only the contribution of NCs located between the dimers (highlighted areas in Figure 6b). On average, 30 pairs of dimers contribute to SERS when irradiated with a laser beam diameter of about 1 μm. In the gaps between the two nanodisks of dimers covered with the NC monolayer (Figure 6b) with a size of 20–30 nm, approximately 17 NCs are placed. Consequently, arrays of gold dimers make it possible to observe SERS from approximately 500 NCs. If we deposit a submonolayer (0.1 monolayer) on top of Au dimers, approximately 50 NCs located in the gaps between the gold dimers contribute to the SERS response, see Figure 9c.

Significant enhancement of the electromagnetic (EM) field in Au dimer arrays and, thus, Raman scattering of NCs on Au dimers is supported by our recent calculations. We developed an approach that makes it possible to calculate the EM field distribution and the LSPR energy for Au dimers of various sizes formed on Si/SiO_2_ substrates [34]. As one can see from Figure 10 for the light polarization parallel to dimer axis, the EM field is predominantly enhanced in the gap between the Au nanodisks in a dimer leading to an enhancement of Raman scattering if an analyte (e.g., CdSe NCs) is deposited on the plasmonic structure. Electrodynamic modeling of nanodimer structures allows figuring out the dependence of the plasmon energy on the structural parameters such as the SiO_2_ layer thickness, the gap size between nanodisks, and predicting the effect of the polarization of the exciting radiation. The study of the numerical model makes it possible to adjust the position of the LSPR in a more comprehensive spectral range to achieve the highest SERS or TERS response.

### 3.4. SERS by CdSe NCs near Single Au Dimers

Since the SERS intensity of the CdSe phonon modes is sufficiently intense, a structure consisting of a single dimer (Figure 2b,c) was used to study the phonon spectrum of several NCs. A NC submonolayer with an amount of NCs 10 times less than in a CdSe NC monolayer was deposited onto a single Au dimer. As shown in Figure 6d, only a few NCs from the sub-monolayer coating are placed in a gap between the nanoclusters in the dimer structure. It should, thus, allow the characteristics of individual NCs, for example, the influence of the NC size, to be investigated. Consequently, a LO mode shift from 208.8 to 207.7 cm^−1^ is observed when measuring the Raman spectrum for two different dimers (Figure 11). This shift can be explained by a CdSe NC size difference of about 1–2 nm, which is within the range of the NC size dispersion of 3–6 nm determined from HRTEM (Figure 5) [48]. The sizes of the NCs contributing to the Raman spectra can be estimated by the phonon confinement model [49]. According to [7], the frequency of the first LO mode decreases with decreasing NC size due to the negative dispersion of LO phonons in CdSe. The frequency difference of the first LO modes for NCs with 6 and 3 nm is 2 cm^−1^ [45]. The experimentally observed value (1.1 cm^−1^) is in this range and the frequency positions of the LO mode are 208.8 and 207.7 cm^−1^ corresponding to CdSe NC sizes of about 5 and 4 nm, respectively [35].

### 3.5. Gap Mode TERS of a CdSe NC Monolayer on Au Nanodisk Arrays

In order to get a further insight into the phonon spectra of CdSe NCs on plasmonic substrates, a TERS map in so-called gap mode was acquired simultaneously with an AFM morphology map. In this mode CdSe NCs are located in the gap between Au nanoclusters of the plasmonic substrate and the metallized (Au) TERS tip. Gap-mode TERS images of CdSe NCs on the Au nanodisk array were obtained for an array period of 150 nm and a size of the Au nanodisks of 100 nm as determined from the AFM image (Figure 12a). The corresponding TERS image created for the CdSe LO mode around 210 cm^−1^ is shown in Figure 12b. As can be seen from Figure 12b, the TERS image represents an array of rings with their diameter and periodicity matching the dimensions of the nanodisks observed in the AFM image. An overlay of TERS and AFM maps clearly shows that the largest TERS signal originates from the edges of the Au nanodisks (Figure 12a,b). This is in accordance with the finite element method calculation of the electric field distribution in the vicinity of an Au nanodisk, which shows the maximal field enhancement near its edges [50].

### 3.6. TERS Imaging of a Single CdSe NC

The computational electrodynamic model made it possible to tune the nanodisk plasmon energy as close as possible to the energies of the CdSe NC exciton and the laser excitation. This approach allows performing TERS mapping and achieving a resonant enhancement of the gap-mode TERS response.

However, there is a significant challenge for TERS mapping in the semicontact mode. During TERS mapping, the cantilever can randomly capture CdSe NC from the sample surface, making the TERS map blurry. The solution to this problem was the use of an intermediate protective layer of a MoS_2_ monolayer between the NC and the cantilevers. The monolayer coating protects the NCs from possible mechanical shear and allows a TERS imaging of a single CdSe NC to be performed.

As a result, we demonstrate TERS imaging of a single CdSe NC located in the vicinity of the Au nanodisk and protected by a MoS_2_ monolayer (Figure 13a) performed at resonant conditions (638.2 nm). Figure 13b,c presents a series of spectra measured along the x and y axes, respectively, with a step of 2 nm. As can be seen from the TERS spectra, when the TERS tip approaches the NC surface along either x or y direction, a gradual increase of the LO mode intensity is observed. Taking into account that TERS spectra were acquired with a lateral resolution of 2.3 nm [37] and the size of CdSe NCs in the experiment was 5–6 nm, one can conclude that Figure 13 refers to Raman scattering response from a single CdSe NC.

## 4. Conclusions

Using the Langmuir–Blodgett technique, monolayers and sub-monolayers of colloidal CdSe NCs were homogeneously deposited on plasmonic substrates designed as arrays of Au nanodisks, dimer arrays, or single dimers. The size and shape of CdSe NCs and Au single nanodisks and dimers were determined from (HR)TEM and SEM experiments. SERS by optical phonons in CdSe NCs on the Au nanodisk arrays was found to be resonantly dependent on the Au nanodisk size and the laser excitation energy. The correlation between size dependence of the LSPR energy of Au nanodisk arrays and SERS enhancement maximum was established. This correlation together with SERS anisotropy observed for Au dimers evidences the electromagnetic enhancement mechanism of the SERS effect. SERS by optical phonons in CdSe NCs deposited on single Au dimers reveals a variation of the phonon peak frequency from one dimer to another that indicates that quasi-single NC phonon spectra are obtained.

SERS and TERS are becoming powerful methods of studying optical phonons of CdSe semiconductor NCs. By adjusting the energy of the laser radiation, the plasmon energy, and the electronic transition in the NC, it is possible to achieve resonance conditions for the SERS and TERS experiments. The methods make it possible to study phonons and determine the NC size of single NCs. Using single dimers, we experimentally demonstrate the shift of the LO CdSe mode due to the size selective Raman scattering for NCs with different sizes. Gap-mode TERS imaging enable us to visualize a single CdSe NC located in the vicinity of Au nanocluster and to deliver information on the NC phonon spectrum. Consequently, SERS and TERS methods allow the detection of low concentrations of material on a nanometer scale, down to a single NC.

## Figures and Tables

**Figure 1 nanomaterials-12-02197-f001:**
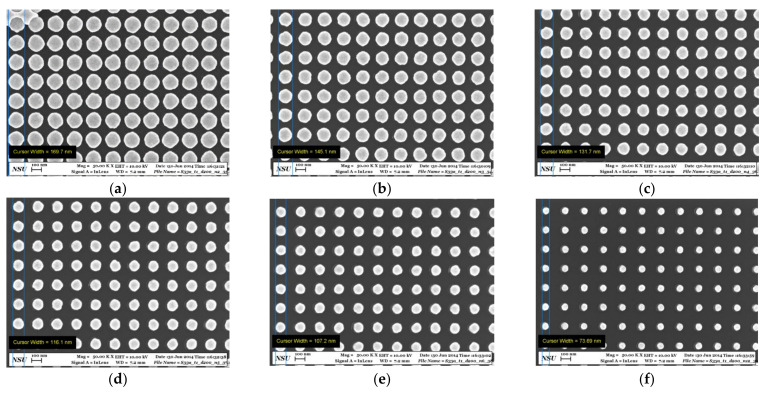
Representative high-resolution SEM images of Au nanocluster arrays with a period of 200 nm and various nanocluster diameters: (**a**) 170 nm, (**b**) 145 nm, (**c**) 132 nm, (**d**) 116 nm, (**e**) 107 nm, and (**f**) 74 nm.

**Figure 2 nanomaterials-12-02197-f002:**
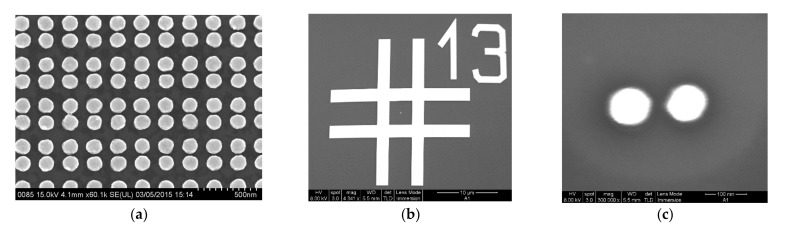
(**a**) Representative high-resolution SEM image of an Au dimer array with a period of 200 nm and a nanodisk diameter of 130 nm. (**b**) SEM image of a single dimer placed in the center of the cross marked as 13. (**c**) Zoomed SEM image of single dimer from (**b**) with nanodisk diameter of 90 nm and a distance (center to center) between nanodisks of 150 nm.

**Figure 3 nanomaterials-12-02197-f003:**
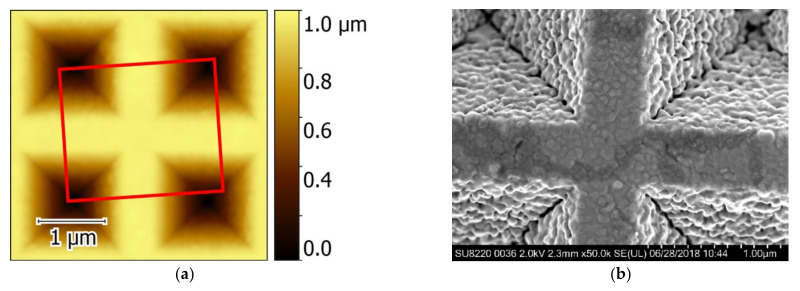
(**a**) AFM image of four inverted pyramids of a Klarite^®^ structure. (**b**) SEM image of the central fragment of the area shown in Figure 1a with a deposited submonolayer of CdSe NCs.

**Figure 4 nanomaterials-12-02197-f004:**
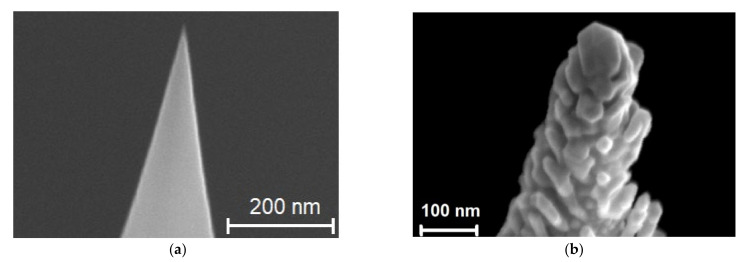
(**a**) SEM image of commercial Si AFM tip (**a**). (**b**) SEM image of TERS tip apex covered with Au/Ti.

**Figure 5 nanomaterials-12-02197-f005:**
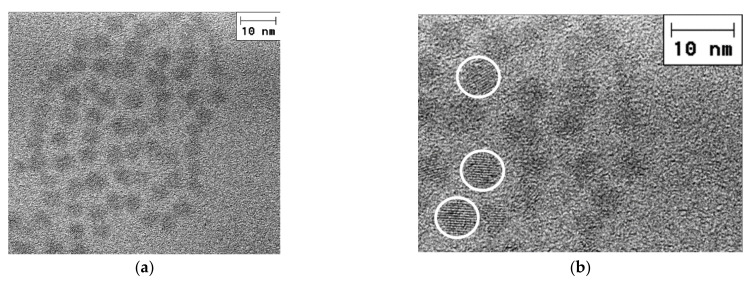
(**a**) HRTEM image of CdSe NCs lying on Si substrate. (**b**) Zoomed area of HRTEM image (**a**) with some NCs highlighted by white circles illustrating the crystalline structure of NCs.

**Figure 6 nanomaterials-12-02197-f006:**
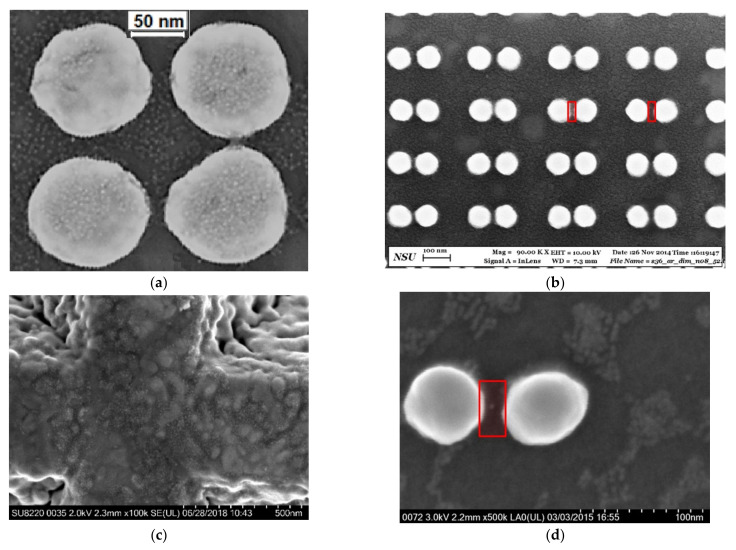
SEM images of a fragment of the Au nanodisk array (**a**), dimers (**b**), and Klarite substrate (**c**) covered homogeneously with CdSe NCs (tiny bright spots). (**d**) Typical SEM image of single nanodimer covered by a submonolayer of CdSe NCs. The red semitransparent rectangular areas indicate regions with CdSe NCs contributing to SERS. The images are adapted from [43]. Copyright 2016, Elsevier.

**Figure 7 nanomaterials-12-02197-f007:**
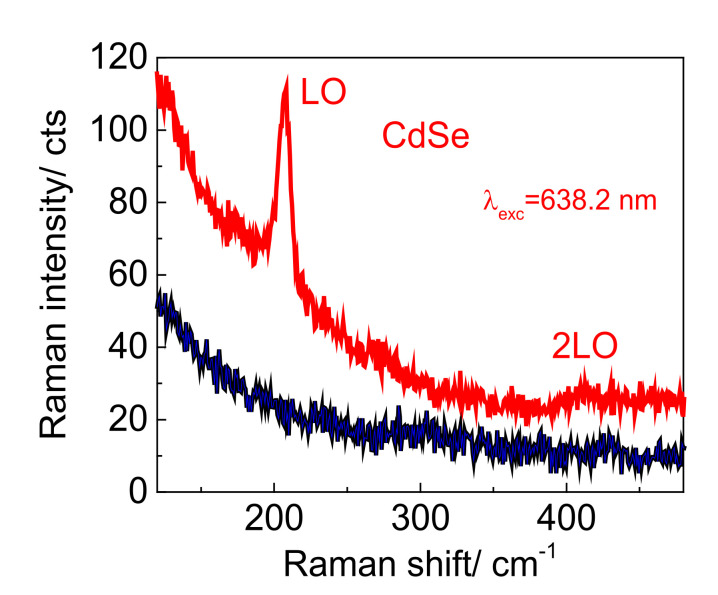
SERS spectra of CdSe NC monolayer measured on a Klarite substrate and on the flat gold, red and blue curves, respectively.

**Figure 8 nanomaterials-12-02197-f008:**
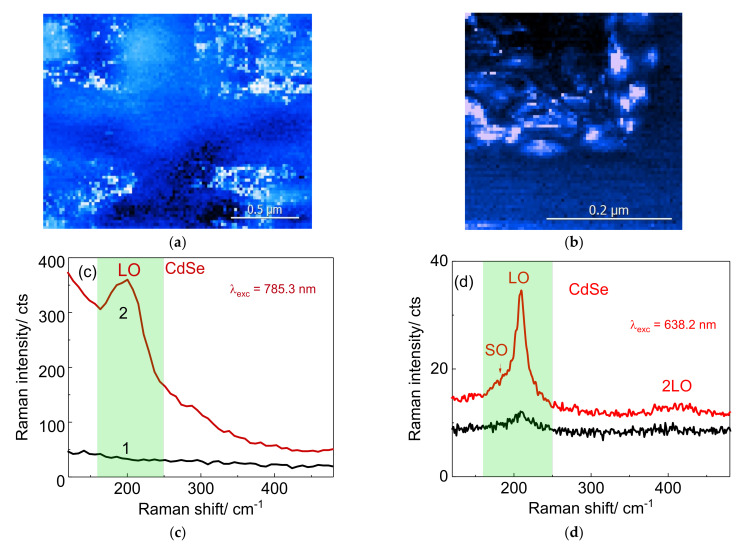
TERS mapping of Klarite structure at a wavelength of (**a**) 785.3 nm and (**b**) 638.2 nm. TERS maps indicate the intensity of the LO CdSe mode depending on the position of the tip during the TERS scan. The most intense TERS spectra, red spectra in (**c**,**d**) are taken from the TERS mapping in (**a**,**b**), measured by using 785.3 and 638.2 nm wavelengths, respectively. The black spectra in (**c**,**d**) were taken on a flat part of the gold surface.

**Figure 9 nanomaterials-12-02197-f009:**
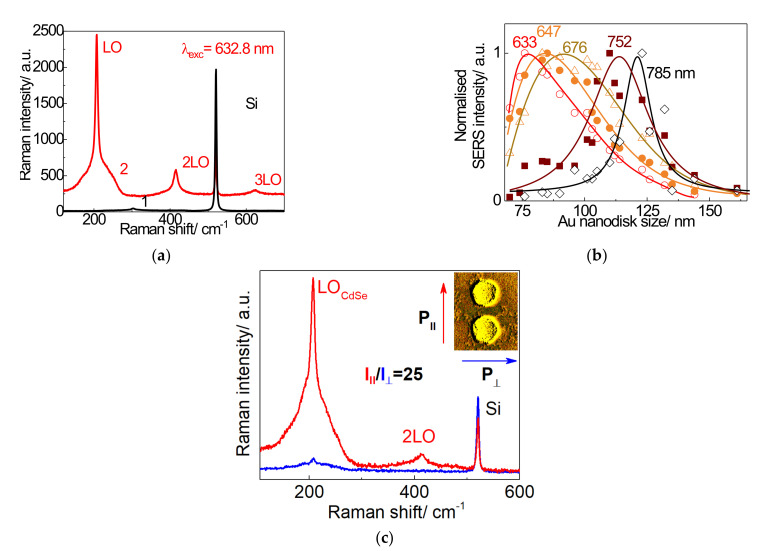
(**a**) Raman spectrum of CdSe NCs on bare Si (black spectrum 1) and resonant SERS spectrum measured on Au nanodisks arrays with diameters of 76 nm and 200 nm pitch between disks (spectrum 2), with an excitation wavelength of 632.8 nm. (**b**) The relative intensity of SERS by LO phonons as a function of Au nanodisk size (SERS profile) measured for 633, 647, 676, 752, and 785 nm excitation wavelengths. Solid lines represent polynomic fits. (**c**) Polarized SERS spectra of CdSe NCs on a dimer array, measured the polarization along (red spectrum) and across (blue spectrum) the dimer axis as shown in the insert. Adapted with permission from [41]. Copyright 2016, Elsevier.

**Figure 10 nanomaterials-12-02197-f010:**
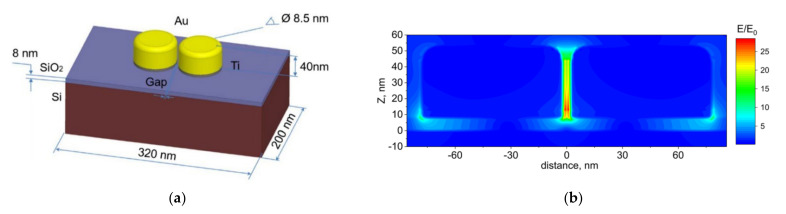
Scheme of the model used to calculate the EM enhancement: (**a**) two gold nanodisks formed on a Si surface with an 8 nm thick SiO_2_ layer. (**b**) Side-view image of the electric field distribution near the two nanodisks. The electric field near the nanodisks was normalized to the incident electric field. Reprinted with permission from [34]. Copyright 2019, Springer.

**Figure 11 nanomaterials-12-02197-f011:**
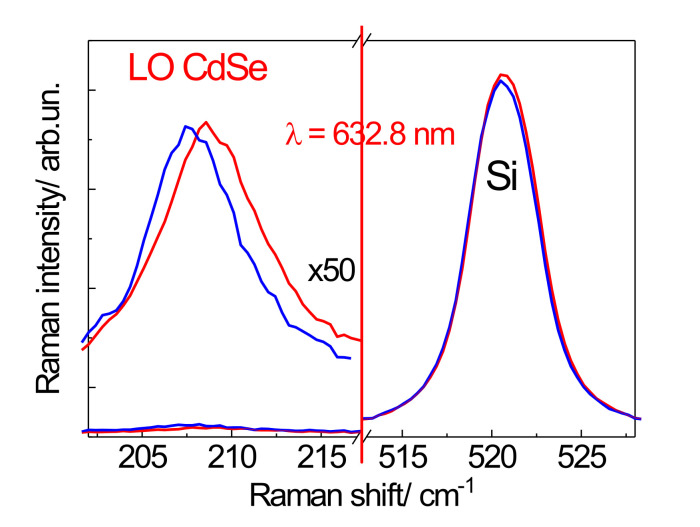
SERS spectra measured on two different single dimers reveal the LO phonon feature around 208 cm^−1^ and a frequency shift of 1.1 cm^−1^. Si peaks are shown for reference. Adapted with permission from [41]. Copyright 2016, Elsevier.

**Figure 12 nanomaterials-12-02197-f012:**
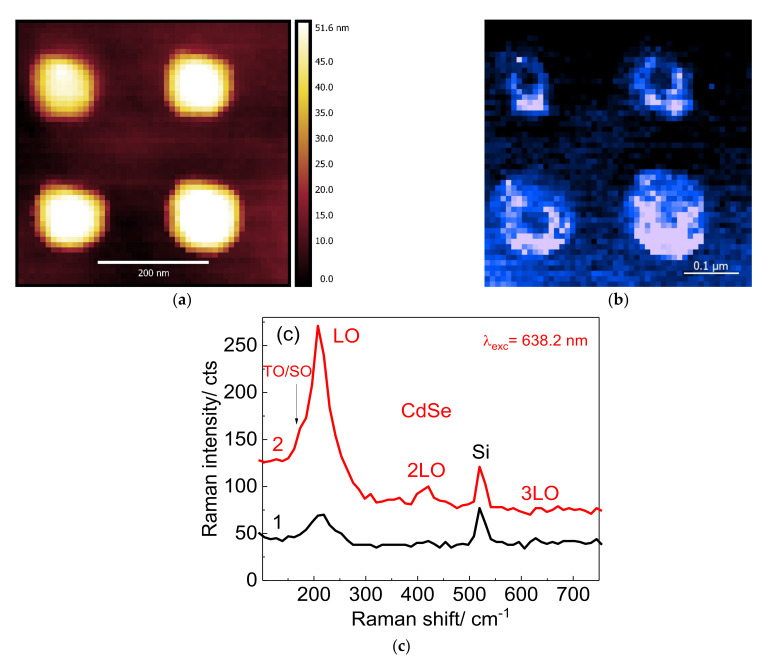
(**a**) AFM image of a nanodisk array covered with a monolayer of CdSe NCs, (**b**) TERS map of LO phonon mode from CdSe NCs measured simultaneously with the AFM image shown in (**a**). The most intense TERS signal in the TERS map comes predominantly from edges of Au nanodisks. (**c**) Comparison of the TERS spectra of CdSe NCs measured on the flat Si/SiO_2_ substrate (black spectrum 1) and at the corner of a nanodisk (red spectrum 2).

**Figure 13 nanomaterials-12-02197-f013:**
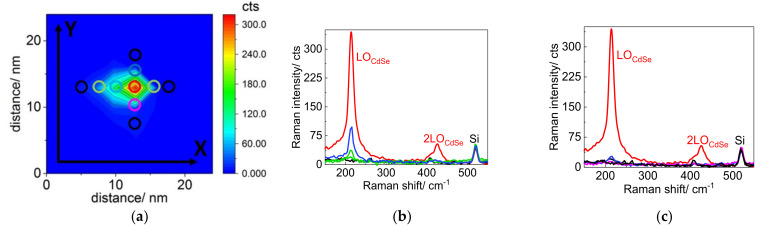
(**a**) Intensity map of the LO CdSe mode of a single NC. Reproduced with permission from [37]. Copyright 2019, Springer. (**a**) TERS map of a single CdSe NCs located in the vicinity of an Au nanocluster. The color of the circles corresponds to the color of the spectra shown in (**b**,**c**). TERS spectra (**b**,**c**) measured with an excitation wavelength of 638.2 nm along the x (from left to right) and y (from bottom to top) axes, respectively. Adapted with permission from [14]. Copyright 2018, IOP Publishing.

## Data Availability

The data that support the findings of this study are available from the corresponding author upon reasonable request.

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
