# Peer review of "Surface- and Tip-Enhanced Raman Scattering by CdSe Nanocrystals on Plasmonic Substrates"

_nanomaterials, 2022, doi:10.3390/nano12132197_

Round 1

Reviewer 1 Report

The paper is a review of the authors' main results on TERS imaging of inorganic NCs. The paper is interesting, well written and well organized. I thus find it overall suitable for publication in Nanomaterials.

I have anyway few comments:

1) The paper basically only contains results from the authors. As inorganic NCs have been widely investigated and as the TERS imaging is made for several systems, it would be interesting to compare the reported results also with other studies on NCs from other groups and on other systems with similar technique. This would allow the readers to properly understand the position on the reported results within the broader words of NCs invesitgation and TERS imaging.

2) A couple of minor points are related to the presence of several warnings related to "missing reference souce", likely evidencing typing error in the text that prevented a correct compilation, and to a badly written y axis label in Fig. 7.

Author Response

First of all, we would like to thank the reviewer for his constructive and useful comments and suggestions. We sincerely appreciate his help. Following his remarks and comments we corrected our manuscript and hope that the present version of the manuscript fits to the quality requirements of the Journal.

Below the answer to the reviewer and the corrections made in the text of the manuscript are given.

Reviewer #1:

The paper is a review of the authors' main results on TERS imaging of inorganic NCs. The paper is interesting, well written and well organized. I thus find it overall suitable for publication in Nanomaterials.

I have anyway few comments:

  • The paper basically only contains results from the authors. As inorganic NCs have been widely investigated and as the TERS imaging is made for several systems, it would be interesting to compare the reported results also with other studies on NCs from other groups and on other systems with similar technique. This would allow the readers to properly understand the position on the reported results within the broader words of NCs investigation and TERS imaging.

Reply to reviewer:

In our review, we focused on SERS and TERS investigations of CdSe NCs and cited the papers devoted to this subject. If we did cite other interesting papers, we would be thankful to the reviewer for giving us the corresponding references. We could include the appropriate discussion in the manuscript.

  • A couple of minor points are related to the presence of several warnings related to "missing reference souce", likely evidencing typing error in the text that prevented a correct compilation, and to a badly written y axis label in Fig. 7.

Reply to reviewer:

We have corrected the references to remove the warnings.

Reviewer 2 Report

In this manuscript, I.A. Milekhin  et. al. presents using the Langmuir-Blodgett technique to investigate monolayers and sub-monolayers of colloi dal CdSe NCs. They investigates the correlation between size dependence of the LSPR energy of Au nanodisk arrays and SERS enhancement maximum. The publication is clear and clearly argued. The conclusions are correct and publication is worthy of Nanomaterials. In this sense, this paper is acceptable provided that the authors addressed the following problems:

1. There are many sentences “Error! Reference source not found” in the whole article. Please revise more carefully. 

2. Figure 6, How to ensure the CdSe NC has formed sub-monolayer as overlap is difficult to avoid?

3. A more clear figs must be presented to replace Figure 10.

Author Response

First of all, we would like to thank the reviewer for his constructive and useful comments and suggestions. We sincerely appreciate his help. Following his remarks and comments we corrected our manuscript and hope that the present version of the manuscript fits to the quality requirements of the Journal.

Below the answer to the reviewer and the corrections made in the text of the manuscript are given.

Reviewer #2:

In this manuscript, I.A. Milekhin  et. al. presents using the Langmuir-Blodgett technique to investigate monolayers and sub-monolayers of colloidal CdSe NCs. They investigate the correlation between size dependence of the LSPR energy of Au nanodisk arrays and SERS enhancement maximum. The publication is clear and clearly argued. The conclusions are correct and publication is worthy of Nanomaterials. In this sense, this paper is acceptable provided that the authors addressed the following problems:

  1. There are many sentences “Error! Reference source not found” in the whole article. Please revise more carefully. 

Reply to reviewer:

Done, we have corrected the references and removed the warnings.

  1. Figure 6, How to ensure the CdSe NC has formed sub-monolayer as overlap is difficult to avoid?

Reply to reviewer:

We agree with reviewer that CdSe NCs can overlap in a submonolayer. However, the average coverage of CdSe NCs represents islands composed of CdSe NCs with an average thickness of about 6 nm distributed on a bare Si surface (Figure 6d).

  1. A more clear figs must be presented to replace Figure 10.

Reply to reviewer:

Figure 10 is replaced by a picture with a better resolution.
